# The Contribution of Phytate-Degrading Enzymes to Chicken-Meat Production

**DOI:** 10.3390/ani13040603

**Published:** 2023-02-09

**Authors:** Peter H. Selle, Shemil P. Macelline, Peter V. Chrystal, Sonia Yun Liu

**Affiliations:** 1Poultry Research Foundation within The University of Sydney, Camden, NSW 2570, Australia; 2Sydney School of Veterinary Science, The University of Sydney, Camden, NSW 2570, Australia; 3School of Life and Environmental Sciences, Faculty of Science, The University of Sydney, Camden, NSW 2570, Australia; 4Complete Feed Solutions, Pakuranga, Auckland 2140, New Zealand

**Keywords:** amino acids, broiler chickens, phosphorus, phytase, phytate, protein

## Abstract

**Simple Summary:**

Dietary inclusions of phytate-degrading enzymes have made immense contributions to sustainable chicken-meat production since the turn of this century. Firstly, exogenous phytases have attenuated the phosphorus (P) pollution of the environment and conserved the globe’s diminishing, finite P reserves by enzymatically dephosphorylating their substrate, phytate. Secondly, phytases have enhanced the growth performance of broiler chickens by counteracting the potent anti-nutritive properties of phytate and/or reducing feed ingredient costs in broiler diets via the application of phytase matrix values to least-cost formulations.

**Abstract:**

The contribution that exogenous phytases have made towards sustainable chicken-meat production over the past two decades has been unequivocally immense. Initially, their acceptance by the global industry was negligible, but today, exogenous phytases are routine additions to broiler diets, very often at elevated inclusion levels. The genesis of this remarkable development is based on the capacity of phytases to enhance phosphorus (P) utilization, thereby reducing P excretion. This was amplified by an expanding appreciation of the powerful anti-nutritive properties of the substrate, phytate (*myo*-inositol hexaphosphate; IP_6_), which is invariably present in all plant-sourced feedstuffs and practical broiler diets. The surprisingly broad spectra of anti-nutritive properties harbored by dietary phytate are counteracted by exogenous phytases via the hydrolysis of phytate and the positive consequences of phytate degradation. Phytases enhance the utilization of minerals, including phosphorus, sodium, and calcium, the protein digestion, and the intestinal uptakes of amino acids and glucose to varying extents. The liberation of phytate-bound phosphorus (P) by phytase is fundamental; however, the impacts of phytase on protein digestion, the intestinal uptakes of amino acids, and the apparent amino acid digestibility coefficients are intriguing and important. Numerous factors are involved, but it appears that phytases have positive impacts on the initiation of protein digestion by pepsin. This extends to promoting the intestinal uptakes of amino acids stemming from the enhanced uptakes of monomeric amino acids via Na^+^-dependent transporters and, arguably more importantly, from the enhanced uptakes of oligopeptides via PepT-1, which is functionally dependent on the Na^+^/H^+^ exchanger, NHE. Our comprehension of the phytate–phytase axis in poultry nutrition has expanded over the past 30 years; this has promoted the extraordinary surge in acceptance of exogenous phytases, coupled with the development of more efficacious preparations in combination with the deflating inclusion costs for exogenous phytases. The purpose of this paper is to review the progress that has been made with phytate-degrading enzymes since their introduction in 1991 and the underlying mechanisms driving their positive contribution to chicken-meat production now and into the future.

## 1. Introduction

The incorporation of inorganic phosphorus (P) supplements, protein, and energy into diets for broiler chickens is a costly but mandatory input. The contribution that phytate-degrading enzymes have made, and will continue to make, to sustainable chicken-meat production is their capacity to enhance the utilization of these vital dietary components, coupled with the environmental advantages. Superficially, the genesis of this contribution is straightforward in that exogenous phytases liberate phytate-bound P and counteract the broad, anti-nutritive properties of dietary phytate or the polyanionic molecule, *myo*-inositol hexaphosphate (IP_6_). Phytate is invariably present in all practical broiler diets; this now applies almost equally to phytases, given their exceptional present level of acceptance by the global chicken-meat industry. 

A tsunami of phytate and phytase research articles and review papers have been published since the landmark Simons et al. [1] study in which a fungal phytase (*Aspergillus ficuum*) at 1000 FTU/kg reduced P excretion by 25.9% (2.0 versus 2.7 g/kg feed intake) in broiler chickens. The reasons for the remarkable increase in the acceptance of exogenous phytases over the last two decades, pursuant to their introduction in 1991, are evident in a series of reviews [2,3,4,5,6,7,8,9,10]. This list is certainly not exhaustive, but the reviews do reflect the advances made in our comprehension of the phytate–phytase axis in poultry nutrition over this timeframe. The objective of this review is to focus on the recent developments driving the positive contributions to chicken-meat production that have been achieved by phytate-degrading feed enzymes, to quantify the positive contributions, and to identify areas where research may amplify these contributions into the future.

## 2. Background

### 2.1. Phytate

Phytate (*myo*-inositol hexaphosphate; IP_6_) was first identified more than 150 years ago [11], and it is present in all feedstuffs of plant origin, predominantly as the magnesium (Mg) and potassium (K) salt of phytate, which may be represented as Mg_3_–K_6_–IP_6_ [12]. Phytate contains 282 g/kg P (phytate-P), but the P component of phytate is inadequately utilized by pigs and poultry, resulting in P excretion. This constitutes an environmental hazard as the P pollution of freshwater reserves leads to algal blooms and eutrophication [13,14]. Even more importantly, Abelson [15] identified a potential phosphorus crisis because global rock P reserves are being depleted and predicted that future generations will ultimately face problems in obtaining enough P to exist. One projection is that rock phosphate production will peak in 2033 at a volume approaching 30 million tonnes [16], although other projections suggest that peak phosphorus will occur decades later [17]. Nevertheless, global rock P reserves are finite, hence the potential crisis. The usage of P as fertilizer (82%) clearly exceeds its usage in animal feeds (7%) in a direct sense [18], but there is a considerably greater indirect P usage by pigs and poultry arising from the use of P fertilizers for the production and harvest of relevant feedstuffs. The capacity of exogenous phytases to liberate P moieties from IP_6_ phytate in a step-wise manner is recognized; it decreases P excretion and permits reduced dietary inclusions of inorganic P sources. For example, De Sousa et al. [19] successfully applied phytase matrix values (g/kg) of 1.50 P coupled with 1.65 Ca, 4.2 crude protein, 0.17 lysine, 0.04 methionine, 0.30 threonine, 0.30 arginine, and 0.22 MJ/kg energy to maize–soy broiler diets. The estimated global feed production for broiler chickens was 351 million tonnes in 2021 [20], and if a phytase matrix value of 1.50 g/kg P was applied to this volume, it would equate to a reduction in inorganic P inclusions in broiler diets approaching 525,000 tonnes. This, in turn, would correspond to a sparing of nearly 3 million tonnes of mono- and/or dicalcium phosphate by the global chicken-meat industry. Moreover, phytase inclusion costs are substantially more economical than the corresponding inclusions of mono- and dicalcium phosphate. Therefore, phytate-degrading enzymes are making tangible contributions to both chicken-meat production and the conservation of the globe’s finite P reserves. 

### 2.2. Phytase

Phytase activity was first detected in rice bran more than a century ago [21]. The intrinsic or ‘plant’ phytase activity in rice bran ranged from 70 to 190 FTU/kg in 16 samples, and rice bran is abundant in phytate, containing an average of 56.2 g/kg phytate or 15.86 g/kg phytate-P [22]. Initial attempts made to develop a commercial phytase feed enzyme were focused on the capacity of phytate to limit calcium (Ca) availability [23]. Indeed, it was proposed by Nelson [24] that Ca requirements for broilers could be calculated from dietary phytate-P concentrations using the following equation:Ca_(%)_ = 0.6 + [phytate-P_(%)_ × 1.1](1)

This equation predicts that for a broiler diet containing 2.5 g/kg phytate-P the appropriate specification would be 8.75 g/kg Ca. However, the first commercial phytase feed enzyme (Natuphos^®^), which was of fungal origin (*Aspergillus niger*), was introduced by the Dutch company, Gist-brocades, in 1991 and was initially evaluated by Schöner et al. [25,26]. These researchers reported that phytase inclusions ranging from 700 to 1050 FTU/kg were equivalent to 1 g P as monocalcium phosphate, depending on the parameter assessed and the age of the birds. In the Simons et al. [1] study, another fungal phytase (*Aspergillus ficuum*) was evaluated in which broiler chickens were offered low P diets (1.5 g/kg nonphytate-P) based on maize, sorghum, and soybean meal. A phytase inclusion of 1000 FTU/kg increased P availability from 49.8% to 62.5%, reduced P excretion by 25.9% (2.0 versus 2.7 g/kg feed intake), and significantly improved weight gain and FCR from 1 to 24 days post-hatch.

However, there has been a subsequent transition from phytases of fungal origin to those of bacterial origin, which have greater efficacy. Very few in vivo comparisons have been published, although Ptak et al. [27] reported that the addition of a bacterial phytase at 500 FTU/kg to a negative control diet generated significantly better responses in weight gain by 4.36% (3161 versus 3029 g/bird) and feed intake by 3.80% (5351 versus 5155 g/bird), from 1 to 42 days post-hatch in comparison to a fungal phytase. Moreover, the bacterial phytase significantly enhanced energy utilization by 0.49 MJ (13.13 versus 12.64 MJ/kg AMEn) relative to the fungal phytase. In an unpublished comparison completed on this campus, a bacterial phytase significantly outperformed a fungal phytase in weight gain (8.74%), feed intake (6.04%), retentions of N (8.60%), P (6.29%), and Ca (6.95%) and supported a numerically improved FCR (2.53%). A telling in vitro comparison of seven different phytases was completed by Menezes-Blackburn et al. [28]. One bacterial phytase had a lower optimum pH range (3.0 versus 4.5–5.5) than the fungal phytase and a far greater phytase activity at pH 3.0 and was more resistant to pepsin activity (97.7% versus 58.0%) than the fungal phytase. Moreover, considerably less bacterial phytase activity was required to achieve either a 50% reduction in IP6 or a 50% increase in the release of inorganic P than from the fungal phytase. In another comparison [29], the residual activity of an *Escherichia coli* phytase (94.6%) following pepsin exposure was clearly superior to that of an *Aspergillus niger* phytase (25.9%). Onyango et al. [30] reported that the residual activity of an *Escherichia coli* phytase exceeded that of a *Peniophora lycii* phytase in the crop (1.61), proventriculus and gizzard (6.44), jejunum (22.2), and ileum (15.2) by substantial factors, as shown in the parentheses.

The principal site of the activity of fungal phytases is in the crop [31]; however, because bacterial phytases are resistant to pepsin activity and have a lower pH activity spectra, their principal site of activity is the gizzard, where digesta retention in this powerful grinding organ would facilitate phytate degradation. A bacterial phytase was shown to completely convert IP_6_ phytate to lesser phytate esters in the gizzard [32]. Phytate esters lesser than IP_6_ and IP_5_ are relatively innocuous because their chelating capacity is diminished. For example, the in vitro chelating capacity of IP_4_ phytate to bind Ca as a mineral-phytate complex at pH 7.0 was 32.3% that of IP_6_ phytate, and the capacity of IP_3_ phytate was only 3.28% that of IP_6_ phytate [33]. 

Practical broiler diets contain in the order of 2.82 g/kg phytate-P or 10.0 g/kg IP_6_ phytate, but the actual level depends on the (variable) phytate contents of the relevant feedstuffs. From first principles, the quantity of P liberated by phytase is the product of the dietary phytate-P concentrations and the extent of the phytate degradation. Initially, a P equivalency value of 1.115 g/kg was assigned to a fungal phytase at a 600 FTU/kg inclusion level. Phytate degradation by fungal phytases was in the order of 45.8% over a review of seven experiments [5], which would translate to a P equivalency of 1.292 g/kg in a broiler diet containing 2.82 g/kg phytate-P. Enzymic hydrolysis involves the step-wise dephosphorylation of the polyanionic IP_6_ phytate molecule to release the lesser phytate esters (IP_5_, IP_4_, IP_3_, IP_2_, IP_1_), and the total hydrolysis of IP_6_ phytate generates six P moieties and inositol. The 1500 FTU/kg inclusion of a bacterial phytase in maize–soy broiler diets was shown by Walk et al. [34] to decrease the concentrations of IP_6_, IP_5_, IP_4_, and IP_3_ in the gizzard by 100%, 81.0%, 26.9%, and 71.4%, respectively, but to increase inositol by 83.5%. This liberation of phytate-P is obviously beneficial, but the provision of inositol may advantage broiler growth performance [35]. The complete hydrolysis of phytate could potentially yield in the order of 2.7 g/kg inositol as there is 273 g/kg inositol in IP_6_ phytate. The inclusion of 1.5 g/kg inositol in broiler diets based on a wheat–maize blend was assessed in Cowieson et al. [36]. In an equivocal outcome, inositol improved FCR from 21 to 42 days post-hatch by 7.14% (1.82 versus 1.96), but depressed FCR by 3.20% (1.29 versus 1.25) in the starter phase to 10 days post-hatch.

The capacity of phytase to liberate phytate-bound P, reduce dietary inclusions of inorganic P supplements such as dicalcium phosphate, preserve finite global P reserves, and attenuate environmental P pollution is vital. However, exogenous phytases have the potential to eliminate inorganic P sources from broiler diets and further their contributions to chicken-meat production. This potential was demonstrated in a series of studies [37,38,39]. Instructively, Marchal et al. [37] estimated that the total removal of inorganic P would reduce the global chicken-meat industry’s annual usage of monocalcium phosphate by 1 million tonnes. The elimination of inorganic P sources from broiler diets may be facilitated by the dietary inclusions of feedstuffs with relatively high phytate contents, including sunflower meal (8.2 g/kg phytate-P) and wheat middlings (8.1 g/kg phytate-P), where the analyzed phytate concentrations were reported in [38].

## 3. Extra-Phosphoric Effects of Exogenous Phytase

The term ‘extra-phosphoric effects’ was probably coined initially by Ravindran [40] in an overview of exogenous phytases in poultry. However, the implication that the benefits of phytase extended beyond the liberation of phytate-bound P was then a hotly debated issue. Nevertheless, the capacity of exogenous phytases to improve amino acid digestibility coefficients is now recognized as their principal extra-phosphoric effect. 

“The significance of the protein–phytate complex in the digestive tract of animals has not yet been determined; whether this is associated with a low absorption of protein as amino acids is by no means certain”. This opinion was expressed by Hill and Tyler [41] nearly seven decades ago; nevertheless, it remained pertinent as it was subsequently contended that phytase does not improve amino acid utilization [42]. This was despite the prior studies [43,44] that demonstrated that exogenous phytase did in fact have a positive impact on amino acid digestibilities. In Ravindran et al. [44], a fungal phytase at 500 FTU/kg increased the average apparent ileal digestibility coefficients of 15 amino acids by 4.34% (0.809 versus 0.775) and by 5.68% (0.819 versus 0.775) at 1000 FTU/kg phytase in birds offered diets based on a wheat–sorghum blend at 28 days post-hatch. 

However, a bacterial phytase at 1000 FTU/kg was subsequently shown to increase average ileal digestibility coefficients of 17 amino acids by 12.30% (0.840 versus 0.748) in birds offered maize–soy diets at 21 days post-hatch [45]. Across the essential amino acids, the responses to phytase ranged from 5.45% (0.928 versus 0.880) for methionine to 15.7% (0.765 versus 0.661) for threonine, which is a typical pattern. More recently, a bacterial phytase at 1000 FTU/kg was reported to increase the average ileal digestibility coefficients of 17 amino acids by 10.30% (0.814 versus 0.738) at 21 days post-hatch [46]. The diets were based on maize, soybean meal, and sunflower meal, with analyzed concentrations of 8.8 g/kg Ca, 4.5 g/kg total P, and 2.82 g/kg phytate-P. Moreover, Dersjant-Li et al. [47] modeled amino acid digestibilities and predicted a response to 2000 FTU/kg phytase of 6.58% (0.81 versus 0.76) in the mean apparent ileal digestibility coefficients of 18 amino acids. A meta-analysis of 24 studies, published from 1996 to 2015, into the effect of phytase on ileal amino acid digestibility in broilers was completed by Cowieson et al. [48]. Overall, phytase improved the digestibilities of 18 amino acids from 0.80 to 0.84. Thus, rather than being uncertain, the evidence for the ‘protein effect’ of phytase is now compelling, and the “low absorption of protein as amino acids” may well be pivotal. 

However, the magnitude of the increases in amino acid digestibilities generated by exogenous phytases do fluctuate, as is evident in Table 1. A comparison between the pronounced responses reported in [45,46], as opposed to the outcomes of the meta-analysis [48] for 17 amino acids, is tabulated. Collectively, there was a combined response of 11.73% (0.819 versus 0.733) in [45,46], but a far more conservative response of 4.13% (0.833 versus 0.800) in the 24 studies assessed in [48]. This three-fold difference in the magnitude of responses certainly deserves a consideration of the mechanisms whereby phytate and phytase influence amino acid digestibilities. 

### 3.1. Mechanisms Underlying the Protein Effect of Phytate and Phytase

The de novo formation of binary protein–phytate complexes at a pH less than the isoelectric point (iP) of protein in the gut is crucial [49]. When the gut pH is less than the iP of protein, the proteins carry a net positive charge and electrostatic attractions between the positively charged proteins, and polyanionic IP_6_ phytate molecules trigger the formation of binary protein–phytate complexes via basic amino acid residues. These complexes have been examined by several research groups [50,51,52,53], and it may be concluded that phytate generates insoluble protein aggregates, which render protein less readily digestible. However, our understanding of protein–phytate interactions, especially on a molecular and structural basis, remains in its infancy [54]. The basic amino acids (arginine, histidine, and lysine) are polar and hydrophilic and are normally found on the outer surface of a protein [55]. Thus, the extent of the protein–phytate interaction may be dependent on the number of unhindered basic amino acid residues in a given protein [56]; nevertheless, the pH is critical. The interactions between sodium phytate and bovine serum albumin were examined in vitro by Prattley et al. [57]. These researchers estimated that 78 basic amino acid binding sites were potentially accessible from a total of 93 in bovine serum albumin. At pH 1, 2, and 3, phytate complexed an average of 42.8% of the protein, but once pH 3.0 was exceeded, the complex formation rapidly declined, but, importantly, Ca had the capacity to disrupt these interactions at a low pH. The extent to which phytate complexes protein in the avian digestive tract is an intriguing question of extreme relevance. From in vitro data [58], it appears that 2.03 mg phytate can bind 21.7 mg soybean meal protein, or about 10.7 times its own weight, at pH 2 and 3. Therefore, it is possible that in a 200 g/kg CP diet containing 10.0 g/kg phytate, phytate could complex in the order of 53.5% of dietary protein, thereby reducing its solubility and digestibility. While this is very much an approximation, it does illustrate the potency of the ‘protein effect’ of phytate and phytase.

The pH of digesta along the digestive tract, especially in the gizzard, in relation to the iP of proteins may be critical to the formation and integrity of binary protein–phytate complexes. In broad-brush terms, the overall iP of ‘protein’ is about 5.7; however, the iP of proteins in feed grains (6.28) is higher than in vegetable protein meals (5.10), as shown in Table 2. As tabulated, the iP of gliadin in wheat is 6.50, as opposed to an iP of 4.70 for glycinin in soybean meal [59], which is a substantial contrast. A pH of 2.60 in gizzard digesta has been recorded in broiler chickens [60], and the real-time pH of digesta in the gizzard of birds offered wheat–soy grower diets containing 500 FTU/kg phytase averaged 2.72 but ranged from 0.91 to 4.74 [61]. While speculative, it may be that the strength or ‘intensity’ of a protein–phytate complex increases with the differential between the actual gizzard pH and the iP of a particular protein. The digesta pH was 4.89 in the crop, 1.98 in the proventriculus, and 3.14 in the gizzard but then rose to 5.53 in the duodenum, 6.06 in the jejunum, and 6.62 in the ileum in birds offered wheat-based diets containing 10.7 g/kg Ca [62]. This is relevant because binary protein–phytate complexes dissociate when the gut pH matches their iPs; therefore, the complexes involving wheat proteins will remain patent longer along the digestive tract than soy proteins. It is then relevant that 1200 FTU/kg phytase improved the average digestibilities of 14 amino acids in wheat by 9.04% (0.844 versus 0.774) as compared to an increase of 4.17% (0.850 versus 0.816) in soybean meal [63]. The more pronounced response in wheat than soybean meal is consistent with the higher iP of wheat than that of soybean meal (Table 2). When gut pH exceeds the iP of protein, the iP proteins become negatively charged; however, ternary protein–phytate complexes may be formed via a cationic bridge, usually Ca^2+^, to link the protein with the phytate. However, the extent to which ternary protein–phytate complexes impede protein digestion is questionable but probably should not be dismissed entirely [6].

Critically, the capacity of pepsin to digest protein bound with phytate in insoluble protein aggregates is compromised; this has been demonstrated in several investigations [64,65,66,67,68,69]. However, this is the result of the phytate-bound protein being rendered refractory to pepsin digestion. The possibility of phytate complexing pepsin is precluded because the amino acid profile of pepsin is almost devoid of basic amino acids [70], and pepsin has an extremely low isoelectric point, with an iP in the order of 1.0 [71]. 

Under in vitro conditions, the phytate inhibition of pepsin activity was maximal at pH 2–3 but negligible at pH 4.0–4.5 [68]. Moreover, Yu et al. [69] concluded that phytate must be degraded from IP_6_ to IP_1–2_ if protein is to be rendered completely vulnerable to pepsin activity, although IP_3–4_ showed some inhibition of pepsin activity but to lesser extents than IP_5–6_. In young pigs, 2.0 g/kg phytate depressed pepsin activity by 46.3% (142.5 versus 265.5 PU/ml) and increased Na jejunal concentrations by 57.0% (4192 versus 2670 ppm) in [72]. In poultry, 500 FTU/kg phytase has been shown to increase pepsin activity in the proventriculus by 9.47% (17.34 versus 15.84 nmol/mg protein) in broilers offered 2.2 g/kg phytate-P, maize–soy diets and by 12.9% (16.76 versus 14.84 nmol/mg protein) when the diets contained 4.4 g/kg phytate-P [73]. Thus, dietary phytate has the potential to compromise the pepsin-driven initiation of the overall protein digestion process; moreover, the peptide end products of pepsin digestion prompt gastrin and cholecystokinin secretions, which play regulatory roles in protein digestion [74,75]. Thus, the digestion of protein in the gut lumen and its regulation is impeded by the de novo formation of binary protein–phytate complexes that are refractory to pepsin activity.

Akin to phytate or phytic acid, the polyphenolic molecule, tannic acid, has the capacity to complex protein [76]. However, tannic acid has been shown to increase the secretion of pepsin and free acidity in the stomachs of rats [77]. These researchers reported that the repeated administration of tannic acid at two concentrations increased the pepsin secretion by either 38.7% (27.6 versus 19.9 mg/mL) or 88.4% (37.5 versus 19.9 mg/mL). Sialic acid is an essential moiety of mucin, and subsequently, Mitjavila et al. [78] reported that dietary tannic acid inclusions significantly increased sialic acid excretion by 37.1% (3.03 versus 2.21 g/rat) in rats. These precedents suggest that the reduced capacity of pepsin to digest phytate-bound protein is triggering compensatory hypersecretions of pepsin and hydrochloric acid (HCl). However, because both pepsin and HCl are ‘internal aggressors’ [79], this prompts protective hypersecretions of mucin and sodium bicarbonate (NaHCO_3_) to counteract these insults. This is supported by the observations that phytase significantly increased the pH of gizzard digesta from 2.88 to 3.28 at 500 FTU/kg and to 3.30 at 2000 FTU/kg in birds offered wheat–soy diets [80]. Essentially, exogenous phytase prevents the formation of binary protein–phytate complexes by the prior hydrolysis of phytate in the crop and gizzard to the advantage of pepsin activity and protein digestion. Therefore, the protective hypersecretions of mucin and NaHCO_3_ are diminished by exogenous phytases as they are effectively made redundant. 

#### 3.1.1. Mucin 

The protein content of porcine ileal mucin is 343 g/kg [81], and the dominant amino acids in avian mucin are threonine (19.0%), serine (15.9%), glutamic acid (9.1%), and proline (8.9%), where the molar proportions are shown in parentheses [82]. Phytate, as Mg_3_–K_6_–IP_6_, increased the mucin excretion from 3.05 to 8.00 g/bird and the sialic acid excretion from 0.30 to 0.49 g/bird in poultry over a 54-h period [83]. Phytate also significantly increased the endogenous losses of threonine, proline, and serine in this study, which is consistent with the mucin amino acid profiles reported by Fang et al. [82]. The administration of 8.5 g/kg phytate increased the total endogenous amino acid flows of 17 amino acids by 43.5% (7455 versus 5194 mg/kg DM intake) in broiler chickens [84]. Moreover, a bacterial phytase (500 FTU/kg) decreased the endogenous amino acid flows by 29.3% (5268 versus 7455 mg/kg DM intake), and the response to a fungal phytase (500 FTU/kg) was less pronounced, with a reduction of 22.2% (5802 versus 7455 mg/kg DM intake) in the Cowieson et al. [84] study. Thus, phytate and phytase reciprocally influence endogenous amino acid flows, which in turn impact the apparent amino acid digestibility coefficients. The total endogenous amino acid flows comprise gastric, pancreatic, hepatic, and intestinal secretions, including the array of digestive enzymes, mucin plus desquamated intestinal epithelial cells [85]. However, endogenous amino acids are re-absorbed to differing extents, and an overall re-absorption rate of 79% was determined in pigs [86], although the corresponding estimates in poultry do not appear to have been made. Amino acids derived from the gut microbiota are not genuinely endogenous, but they are present in distal ileal digesta. It was estimated that 24.1% of the amino acids in distal ileal digesta were of microbial origin, as opposed to 47.8% of dietary origin and 25.1% from endogenous secretions in birds offered 195 g/kg CP, maize-based diets [87]. A complicating factor is that the constituency of the gut microflora in the crop and ileum of broilers may be influenced by exogenous phytase [88]. However, in essence, the extent to which phytases ameliorate mucin secretion in the upper digestive tract will positively influence the apparent amino acid digestibilities because the glycosylated protein of mucin remains largely undigested in the small intestine [89]. 

#### 3.1.2. Sodium 

That phytase has a ‘sodium sparing’ effect in poultry was first reported by Cowieson et al. [90]; these researchers found that dietary IP_6_ phytate and exogenous phytase had reciprocal impacts on endogenous Na losses on a total tract basis. Subsequently, a bacterial phytase at 1000 FTU/kg increased the apparent sodium (Na) ileal digestibility coefficients by 65.6% (−0.177 versus −0.515) as a main effect in birds offered maize–soy diets, in which the phytate levels ranged from 10.0 to 11.8 and 13.6 g/kg [91]. The same phytase, at 500 FTU/kg, increased ileal Na digestibility by 92.3% (−0.04 versus −0.52) in broiler diets formulated to contain 1.80 g/kg Na in [92]. In association with increased Na digestibility, phytase increased the mean apparent ileal digestibility coefficients of 17 amino acids by 5.36% (0.806 versus 0.765), with significant increases observed for 13 individual amino acids. 

Apparent Na digestibility coefficients were determined in four small intestinal segments in broiler chickens in a series of three studies [93,94,95]. These studies included maize-, sorghum- and wheat-based diets and a range of phytase inclusion rates (500, 1000, and 2000 FTU/kg). Collectively, phytase increased the Na digestibility coefficients by 30.5% (−2.443 versus −3.517) in the proximal jejunum; 30.2% (−1.425 versus −2.043) in the distal jejunum; 42.7% (−0.839 versus −1.465) in the proximal ileum; and 25.5% (−0.327 versus −0.439) in the distal ileum at 28 days post-hatch. Clearly, copious quantities of endogenous Na secretions are not being reabsorbed along the small intestine in broiler chickens offered non-phytase-supplemented diets, as evidenced by the highly negative Na digestibility coefficients. This is in accordance with Van der Klis et al. [96]; these researchers concluded that Na secretion exceeds its absorption in the anterior small intestine to a large extent, but unlike the other minerals investigated (K, Ca, Mg), the uptake of Na does occur in the large intestine. Increased endogenous Na secretions to buffer HCl are mainly generated by the pancreas as NaHCO_3_ [97]. Thus, the ‘sodium-sparing’ effect of phytase in broiler chickens is unequivocal, the implications of which may be complex. 

#### 3.1.3. Amino Acid Absorption 

The intestinal uptakes of amino acids along the small intestine may hold more importance than the digestion of protein in the gut lumen because the nutrient absorption is probably rate-limiting on broiler growth performance [98]. In a pivotal study, the effects of 500 FTU/kg phytase in maize-based broiler diets on the digestibilities of amino acids and sodium were determined by Truong et al. [94]. Phytase increased the average apparent digestibility coefficients of 16 amino acids by 49.7% (0.720 versus 0.481) in the proximal jejunum; 20.2% (0.802 versus 0.667) in the distal jejunum; 9.07% (0.878 versus 0.805) in the proximal ileum; and 7.24% (0.904 versus 0.843) in the distal ileum. These unequivocal amino acid responses to phytase are detailed in Table 3. However, phytase also increased Na digestibility by 43.4% (−2.008 versus −3.547) in the proximal jejunum; 34.1% (−1.551 versus 2.352) in the distal jejunum; 47.8% (−0.930 versus −1.780) in the proximal ileum; and 36.1% (−0.568 versus −0.889) in the distal ileum. Moreover, the Na and amino acid digestibility coefficients were linearly related in the proximal jejunum (r = 0.582; *p* = 0.047); the distal jejunum (r = 0.721; *p* = 0.008); the proximal ileum (r = 0.862; *p* < 0.001); and the distal ileum (r = 0.825; *p* = 0.002). These outcomes indicate that the intestinal uptakes of Na and amino acids are interconnected. There are two main transport pathways responsible for Na uptakes along the small intestine; the first is the co-absorption of Na with either glucose or amino acids by Na^+^-dependent transporters, and the second pathway is via the Na^+^/H^+^ exchanger, NHE [99]. However, the pathways for small intestinal uptakes of amino acids are more complicated.

#### 3.1.4. Monomeric Amino Acid Co-Absorption with Sodium Driven by Na⁺/K⁺-ATPase 

The co-absorption of Na with glucose predominantly takes place via the Na^+^-dependent transporter SGLT−1 [100]. However, the co-absorption of Na with monomeric amino acids is conducted by an array of Na^+^-dependent nutrient transporters with overlapping affinities for amino acids, although monomeric amino acids also may be absorbed via Na^+^-independent transporters [101,102]. The Na^+^-dependent intestinal uptakes of Na, glucose, and amino acids are driven by the activity of Na⁺/K⁺-ATPase, or the ‘sodium pump’, which maintains an electrochemical gradient across enterocytes [103,104].

However, there is evidence that phytate and phytase have reciprocal impacts on the functional capacity of Na⁺/K⁺-ATPase. Phytate extracted from sweet potato depressed Na⁺/K⁺-ATPase activity by approximately 68% in the anterior small intestines of rats [105]. Reciprocally, Liu et al. [106] reported that phytase increased sodium pump and glucose concentrations in the small intestinal mucosa in chickens offered maize–soy diets. The inclusion of 1000 FTU/kg phytase in diets containing 7.80 g/kg phytate increased the Na⁺/K⁺-ATPase concentrations in the duodenal mucosa by 17.6% (10.01 versus 8.51 µmol/mg) and by 18.4% (13.59 versus 11.48 µmol/mg) in the jejunal mucosa. In diets containing 15.60 g/kg phytate, phytase inclusions increased the sodium pump concentrations by 19.6% (9.52 versus 7.96 µmol/mg) and 17.2% (12.94 versus 11.04 µmol/mg) in the duodenum and jejunum, respectively. Corresponding increases in the glucose concentrations were observed. 

The underlying mechanisms whereby phytate and phytase reciprocally influence sodium pump activity have yet to be properly identified. Sodium pump activity is largely dependent on cytoplasmic Na concentrations within enterocytes [107]. Thus, a straightforward explanation is that if phytate depletes Na concentrations in the gut mucosa by partitioning Na into endogenous pancreatic NaHCO_3_ secretions, declining Na concentrations in the enterocytes could compromise sodium pump activity to the detriment of the intestinal uptakes of glucose and some amino acids. Reciprocally, phytase attenuates endogenous Na secretions to restore Na concentrations in the enterocytes and enhance the intestinal uptakes of nutrients driven by the sodium pump. However, the mechanisms of sodium pump regulation are complex and include the re-phosphorylation of Na⁺/K⁺-ATPase [108]. Presumably, sodium pump activity would be facilitated by increases in the P availability generated by exogenous phytases for re-phosphorylation, and this mechanism was suggested by Martinez-Amezcua et al. [109] as a possible mode of action for enhanced sodium pump activity in response to dietary phytase inclusions. 

#### 3.1.5. Di- and Tripeptide (Oligopeptide) Absorption via PepT-1 and NHE

The second major pathway for small intestinal Na uptakes is via the sodium-hydrogen exchanger, NHE [99]; the avian exchanger NHE functionally resembles NHE-3 in mammalian species [110]. Amino acids are mainly absorbed as di- and tripeptides or oligopeptides and, unlike glucose, the intestinal uptakes of amino acids are not almost totally reliant on Na^+^-dependent transporters. Indeed, Krehbiel and Matthews [111] asserted that 70 to 85% of amino acids are absorbed as oligopeptides and that the intestinal uptakes of oligopeptides are rapid and energetically more efficient than the uptakes of monomeric amino acids [112]. The intestinal uptakes of oligopeptides principally take place via the peptide transporter PepT-1, a system that has been identified in chickens [113,114]. PepT-1 is located in the intestinal brush border membrane and is an H^+^-coupled co-transporter that requires proton binding to absorb dipeptides and tripeptides [115]. While Pept-1 is not directly Na^+^-dependent, it is nevertheless the Na^+^/H^+^ exchanger, NHE, that generates the proton gradient for H^+^/peptide co-transport via PepT-1 [99]. Therefore, the functionalities of PepT-1 and NHE are inextricably related, as reviewed by Spanier [116]. Elevated dietary protein levels have been shown by Osmanyan et al. to increase the mRNA abundance for Pept-1 by 28.2% (0.91 versus 0.71 × 10^−3^) in poultry [117]. Moreover, a natural dipeptide, glycine–glutamine, was shown to increase the PepT1 mRNA levels by a three-fold factor in the apical membrane of Caco-2 cells [118], and subsequently, a synthetic dipeptide, glycl-l-glutamine, was shown to up-regulate the PepT-1 gene expression in cultured chicken intestinal epithelial cells [119]. The relevance of increased substrate levels up-regulating PepT-1 activity will become evident later in this review.

#### 3.1.6. Digesta pH along the Small Intestine 

Digesta pH along the small intestine probably influences PepT-1 activity and the uptake of oligopeptides. The PepT-1-mediated uptakes of four dipeptides were maximal from 5.2 to 6.2 pH, and apparent affinities for these dipeptides were highly pH-dependent in an investigation of the pH dependence of PepT-1 activity in rabbits [120]. Subsequently, Kennedy et al. [121] reported that functional NHE-3 activity was required for the optimal absorption of dipeptides by PepT-1 and was dependent on pH, extracellular Na^+^, and the maintenance of the trans-membrane H^+^ electrochemical gradient. These researchers concluded that both PepT-1 and NHE-3 operate sub-optimally at the typical mucosal surface pH values of pH 6.1 to 6.8 and that lower pH values would advantage their functionality. 

The pH of digesta in birds offered standard, maize–soy diets ranged from 6.01 to 6.15 in the jejunum and from 5.95 to 7.11 in the ileum [122]. As noted, similar findings were recorded earlier by Shafey et al. [57] in birds offered diets containing 10.7 g/kg Ca. However, the increase in dietary Ca from 10.7 to 25.3 g/kg significantly increased the ileal digesta pH from 6.62 to 7.39 in this study. Given the above, this increase in ileal digesta pH would probably depress PepT-1 and NHE activity; the increased gut pH was achieved by increased dietary inclusions of limestone and dicalcium phosphate. Limestone has a particularly high acid-binding capacity (ABC) of 15,044 mEq/kg at pH 3.0, and the ABC of dicalcium phosphate is also high at 5,666 mEq/kg [123], which accounts for the increase in ileal digesta pH reported in [57]. The variations in ABC reported in [123] are noticeable as the coefficients of variation were 14.1% in 13 limestone samples and 32.7% in 5 samples of dicalcium phosphate. Thus, the inclusion of phytase in broiler diets, coupled with the assignation of matrix values for Ca and P, resulting in lower limestone and dicalcium phosphate inclusions, should reduce both the dietary ABC and the digesta pH along the small intestine to the advantage of PepT-1 and NHE activity. 

In the amino acid digestibility assay by Amerah et al. [45], limestone inclusions were increased in a stepwise manner from 3.0 to 8.3, 13.5, and 18.7 g/kg in otherwise identical, maize–soy diets. The limestone increases per se linearly (*p* = 0.009) depressed the mean apparent ileal digestibility coefficients of 17 amino acids by up to 7.95% (0.718 versus 0.780). Similarly, the amino acid digestibilities were assessed in diets containing limestone inclusions of either 5.6 g/kg or 8.2 g/kg in [124]. The transition to the higher limestone level depressed the mean digestibility of 17 amino acids by 4.02% (0.765 versus 0.797). Both outcomes are consistent with the high ABC of limestone elevating the pH along the small intestine, depressing PepT-1 activity, and compromising the intestinal uptakes of oligopeptides. 

In addition, the ABC (mEq/kg) of feed ingredients at pH 4.0 was determined in [125] with limestone at 18,384, soybean meal at 602, and maize at 84. Moreover, six non-bound amino acids had a mean ABC of 149 mEq/kg, which ranged from 83 (*l*-lysine) to 200 (*l*-isoleucine). These data are relevant to the phytase inclusions in reduced-CP diets where the partial replacement of soybean meal with maize coupled with non-bound amino acid inclusions will reduce the dietary ABC, as discussed later.

#### 3.1.7. Rationale for a Novel Postulate 

The genesis of pronounced amino acid digestibility responses to exogenous phytase inclusions reported in Amerah et al. [45], Martínez-Vallespín et al. [46], and Truong et al. [94] could involve the enhanced functionality of the related PepT-1 and NHE transport systems because amino acids are absorbed mostly as oligopeptides. The regulation of the intestinal peptide transporter PepT-1 is complex, but the prime factor influencing the function and/or expression of PepT-1 is its own substrates [126]. Therefore, key to this postulate is the real possibility that exogenous phytases, by rendering more dietary protein vulnerable to pepsin activity, are facilitating the conversion of polypeptides to oligopeptides along the small intestine, thereby up-regulating PepT-1 activity. Support is provided in Moss et al. [127] as phytase inclusions of 750 and 1000 FTU/kg were associated with average numerical increases in ileal mRNA Pept-1 expression from 0.621 to 1.361 and 1.411 in birds offered 195 g/kg CP diets. A schematic representation is presented in Figure 1, in which the intestinal uptakes of oligopeptides, monomeric amino acids, sodium, and glucose via PepT-1, NHE, Na^+^-dependent amino acid transporters, and SGLT-1 are depicted. Pepsin initiates protein digestion by degrading polypeptides; therefore, phytase, by effectively facilitating pepsin activity, should accelerate the conversion of polypeptides to di- and tripeptides. Increased oligopeptide concentrations in the small intestinal digesta should up-regulate Pept-1 activity, based on the mRNA abundance data and promote the intestinal uptakes of oligopeptides to the advantage of amino acid digestibilities. Concurrently, the intestinal uptakes of monomeric amino acids by co-absorption with glucose are enhanced via their appropriate Na^+^-independent transporters, driven by increased Na pump activity located in the baso-lateral membrane of the enterocytes. However, the likelihood is that substantially more amino acids are absorbed as oligopeptides than as monomeric entities. Phytase unequivocally improves Na digestibility, probably largely because the endogenous Na secretions are diminished. Nevertheless, some of the Na digestibility increases may stem from enhanced intestinal Na uptakes via NHE directly or via co-absorption with glucose via SGLT-1 and/or with amino acids via the relevant Na^+^-dependent transporters. In theory, any increases in intestinal Na uptakes via NHE would favor PepT-1 function by proton donation [116]. The reduced limestone and dicalcium phosphate inclusions in phytase-supplemented diets should generate lower dietary acid-binding capacities and reduce gut pH, to the benefit of PepT-1 activity. Thus, the more pronounced phytase-generated amino acid digestibility responses may reflect the enhanced absorption of oligopeptides via PepT-1. It may be possible to test this hypothesis by size exclusion chromatography [128] to determine the effect of exogenous phytase on the peptide molecular weights in digesta along the digestive tract. 

### 3.2. Energy Effect of Phytase

The energy effect of phytase is less consistent than the protein effect. For example, in one comparative study a bacterial phytase (1000 FTU/kg) significantly improved AME in maize-based diets by 0.58 MJ (12.82 versus 12.24 MJ/kg), but numerically depressed AME in sorghum- (12.16 versus 12.30 MJ/kg) and wheat-based (12.05 versus 12.19 MJ/kg) diets [129]. Phytase increased AME (or AMEn) by an average of 0.36 MJ (13.64 13.27 MJ/kg) across a survey of 12 studies where responses range from −0.10 to 0.70 MJ [5]. Phytase increased the proximal ileal starch digestibility coefficients overall by 2.56% (0.920 versus 0.897; *p* = 0.022) in the Liu et al. [129] study but did not influence starch digestibility in three other small intestinal segments. Phytases have been shown to improve starch digestibility [130] and more recently in [131], where 500 FTU/kg phytase increased the starch digestibility coefficients by 17.6% (0.774 versus 0.658; *p* = 0.002) in the proximal jejunum; by 4.80% (0.873 versus 0.833; *p* = 0.062) in the distal jejunum; by 4.72% (0.932 versus 0.890; *p* = 0.062) in the proximal ileum; and by 2.51% (0.938 versus 0.915; *p* = 0.155) in the distal ileum. Starch digestibility is inherently high, particularly in maize starch [132], which leaves little scope for phytase to improve the digestibility coefficients. The likelihood is that any phytase-induced improvements in starch digestibility coefficients stem from the enhanced intestinal uptakes of glucose via the Na^+^-dependent transporter SGLT-1 rather than enhanced starch digestion in the gut lumen. 

## 4. Phytase Matrix Values 

The contribution that phytate-degrading enzymes have made to chicken-meat production may be expressed in economic terms from the assessments of phytase matrix values. The application of phytase matrix values for a range of nutrients in the least-cost formulation of broiler diets is common practice [133]. Phytase suppliers have developed phytase matrix recommendations, and chicken-meat producers may assign matrix values for P, Ca, and Na with the option to include amino acids and energy. This is complicated as some producers may elect to elevate phytase inclusion rates above the standard 500 FTU/kg, or the so-called practice of ‘super-dosing’ [134], and assign more generous matrix values to selected nutrients. Phytase matrix values ranging from 1.50, 1.95 to 2.25 g/kg for available P; 1.65, 2.15 to 2.45 g/kg for Ca; and 0.35, 0.45 to 0.53 g/kg for Na for the corresponding phytase inclusions of 500, 1000, and 1500 FTU/kg, respectively, were evaluated by De Freitas et al. [135]. Phytase inclusions compensated for these mineral reductions as no changes in growth performance, carcass traits, tibial ash, and P contents were observed; indeed, the 1500 FTU/kg phytase inclusion numerically improved weight gain (2886 versus 2785 g/bird) and FCR (1.62 versus 1.65) from 1 to 42 days post-hatch. Integrated chicken-meat producers may choose to confine the phytase matrix to P, Ca, and Na and accept the positive impacts of phytase on protein/amino acids and energy as enhanced growth performance. Alternatively, 1000 FTU/kg phytase matrix values were assigned to starter, grower finisher, and withdrawal diets for P, Na, Ca, crude protein, eleven amino acids, and energy in Moss et al. [136], as shown in Table 4. Birds offered phytase-supplemented diets exhibited similar or improved growth performance in comparison to their counterparts offered control diets; thus, phytase accommodated the assigned matrix values. The diets were based on maize, wheat, soybean meal, lupin, and canola seed, and the control diets were formulated to Cobb nutrient recommendations. On this basis, and taking current (November, 2022) Australian feed ingredient prices into account, the phytase matrix values reduced estimated feed ingredient costs by an average of AUD 65.47 per tonne, as shown in Table 5. This equates (0.675 exchange rate) to a saving of USD 44.20 per tonne. If this were to be applied to the total global feed volume of 351 million tonnes of diets for broiler chickens, this would be equal to a saving of USD 15.5 billion in feed ingredient costs. This estimate is inflated by generous matrix values and high Australian feed ingredient costs; nevertheless, it provides an indication of the magnitude of the tangible contributions exogenous phytases could potentially make to global chicken-meat production.

## 5. Future Directions

At a fundamental level, investigations into the impacts of dietary phytate and exogenous phytase on the intestinal uptakes of di- and tripeptides via the oligopeptide transporter PepT-1 should be completed. As mentioned, size exclusion chromatography to determine the molecular weights of peptides in digesta along the digestive tract should be instructive as it is possible that phytases will facilitate the conversion of polypeptides into oligopeptides by enhancing pepsin activity. Equally, the intestinal uptakes of monomeric amino acids via Na^+^-dependent transporters should be investigated where the reciprocal impacts of phytate and phytase on Na⁺/K⁺-ATPase activity appear to be important but require clarification.

At an applied level, there is considerable interest in the development of reduced-crude protein (CP) diets because of the several potential advantages they hold for chicken-meat production [137,138,139]. The advantages are not limited to, but do include, a diminished dependence on imported soybean meal for most countries in the world simply because synthetic and crystalline amino acids are effectively alternative ‘protein’ sources to soybean meal [140]. However, reduced-CP diets possess lower concentrations of phytate because of the partial replacement of soybean meal with maize and/or wheat. In a series of four similar studies [141,142,143,144], analyzed dietary CP was reduced by an average of up to 47 g/kg (166 versus 213 g/kg CP) following increases in maize (556 to 732 g/kg) inclusions at the expense of soybean meal (157 versus 334 g/kg). However, these dietary manipulations reduced the estimated phytate concentrations by 16.0% from 9.54 to 8.01 g/kg. It is reasonable to expect that lower dietary phytate concentrations would depress the responses to exogenous phytases [43,145]. Alternatively, it may be deduced from the same four studies that the phytate:intact protein ratio increased from 0.046 to 0.057, which could promote binary protein–phytate complex formation. However, the addition of 600 FTU/kg fungal phytase to 216 g/kg CP starter and 170 g/kg CP grower wheat-based diets depressed weight gain and feed intake without influencing FCR from 1 to 42 days post-hatch in [146]. However, phytase inclusions (0, 750, 1500 FTU/kg) in association with pre-pellet cracked maize were shown to improve the growth performance of broilers offered 220, 195, and 170 g/kg CP diets in a Box–Behnken designed study [129]. More promising outcomes to a bacterial phytase at 1500 FTU/kg were observed by Hofmann et al. [147], but the CP reductions evaluated were moderate in this study. Reduced dietary inclusions of limestone and mono- or dicalcium phosphate in association with phytase additions will lower the ABC of broiler diets and the pH along the digestive tract, which may advantage the intestinal uptakes of oligopeptides. Thus, further investigations into this aspect are justified.

Dietary CP reductions generate perturbations in apparent amino acid digestibilities [148]; this is not surprising because two opposing forces are at play. On one hand, reduced dietary protein concentrations depress the apparent digestibility coefficients because concentrations of dietary amino acids in the distal ileum are diluted by amino acids of endogenous and microbial origins [149]. Moreover, the amino acid digestibilities of soybean meal are superior to those of maize, and wheat [150]; so, the partial replacement of soybean meal will tend to depress the amino acid digestibilities in reduced-CP diets. On the other hand, synthetic and crystalline amino acids are notionally completely digestible [151], and their dietary inclusions tend to increase their digestibilities. Phytate has the potential to bind synthetic or crystalline amino acids, depending on their iPs, and the phytate in rice bran has been shown to bind lysine HCl in vitro [152]. Thus, perturbations in amino acid digestibilities may be compounded further by phytase inclusions. Clearly, any clarification as to intestinal amino acid uptakes in poultry should benefit our comprehension of the protein effect of phytate and phytase in general and particularly in the context of reduced-CP diets.

## Figures and Tables

**Figure 1 animals-13-00603-f001:**
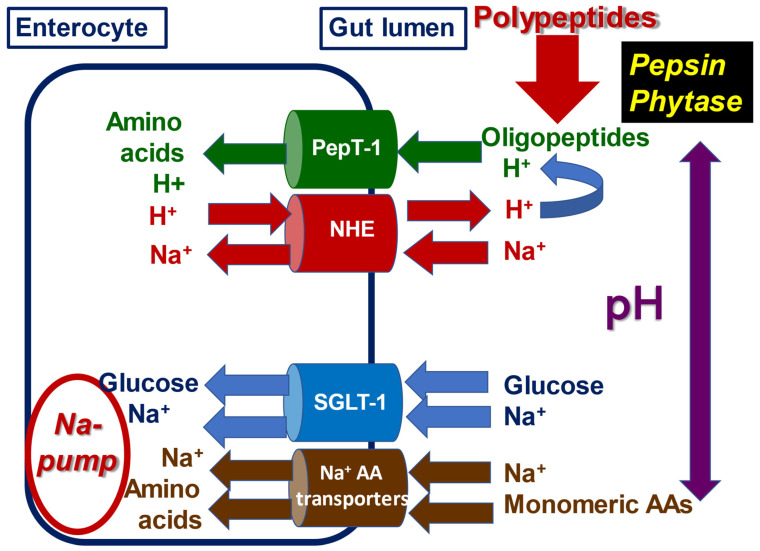
Schematic representation of intestinal uptakes of oligopeptides, glucose, sodium, and monomeric amino acids via four transport systems: PepT-1, NHE, SGLT-1, and Na^+^-dependent amino acid transporters. Adapted in part from Spanier [116]. Note: digesta pH along the small intestine influences the integrity of binary protein–phytate complexes and the pepsin-mediated conversion of polypeptides to oligopeptides plus the functionality of the related transport systems of PepT-1 and NHE.

**Table 1 animals-13-00603-t001:** Effects of exogenous phytase on apparent ileal digestibility coefficients of 17 amino acids in one meta-analysis [48] and two individual assays [46,47].

Amino Acid	Cowieson et al. [48]			Martinez-Vallespin et al. [47]			Amerah et al. [46]		
	Nil	Plus	Response (%)	Nil	1000 FTU/kg	Response (%)	Nil	1000 FTU/kg	Response (%)
Arginine	0.86	0.89	3.49	0.859	0.899	4.66	0.823	0.900	9.36
Histidine	0.80	0.84	5.00	0.725	0.841	16.00	0.730	0.821	12.47
Isoleucine	0.79	0.85	7.59	0.743	0.794	6.86	0.721	0.821	13.87
Leucine	0.82	0.85	3.66	0.749	0.826	10.28	0.722	0.826	14.40
Lysine	0.83	0.86	3.61	0.804	0.862	7.21	0.806	0.889	10.30
Methionine	0.89	0.90	1.12	0.768	0.883	14.97	0.880	0.928	5.45
Phenylalanine	0.82	0.85	3.66	0.806	0.858	6.45	0.737	0.834	13.16
Threonine	0.73	0.77	5.48	0.646	0.752	16.41	0.661	0.765	15.73
Valine	0.78	0.82	5.13	0.689	0.730	5.95	0.669	0.774	15.70
Alanine	0.80	0.83	3.75	0.731	0.818	11.90	0.699	0.807	15.45
Aspartic acid	0.79	0.82	3.80	0.796	0.835	4.90	0.756	0.850	12.43
Cysteine	0.68	0.72	5.88	0.478	0.681	42.47	0.543	0.685	26.15
Glutamic acid	0.87	0.90	3.45	0.833	0.888	6.60	0.807	0.882	9.29
Glycine	0.76	0.79	3.95	0.636	0.749	17.77	0.658	0.765	16.26
Proline	0.80	0.83	3.75	0.789	0.809	2.53	0.722	0.812	12.47
Serine	0.78	0.81	3.85	0.723	0.805	11.34	0.684	0.809	18.27
Tyrosine	0.80	0.83	3.75	0.767	0.813	6.00	0.755	0.844	11.79
Mean	0.800	0.833	4.13	0.738	0.814	10.30	0.728	0.824	13.19

**Table 2 animals-13-00603-t002:** The isoelectric point (iP) of various proteins: selected from Csonka et al. [59].

Feed Grains		Vegetable Protein Meals	
Source	iP	Source	iP
Maize–zein	6.2	Cottonseed–α-globulin	5.5
Rye–gliadin	6.6	Cottonseed–ß-globulin	5.4
Sorghum–kafirin	5.9	Navy bean–phaseolin	4.5
Wheat–gliadin	6.5	Peanut–arachnin	5.4
Wheat bran–prolamin	6.2	Soybean–glycinin	4.7
Mean	6.28	Mean	5.10

**Table 3 animals-13-00603-t003:** Effects of 500 FTU/kg phytase on apparent amino acid digestibility coefficients in four small intestinal segments [adapted from Truong et al. [94].

Amino Acid	Proximal Jejunum			Distal Jejunum			Proximal Ileum			Distal Ileum		
	0 FTU/kg	500 FTU	Response	0 FTU/kg	500 FTU	Response	0 FTU/kg	500 FTU	Response	0 FTU/kg	500 FTU	Response
Arginine	0.609	0.798	31.0%	0.770	0.869	12.9%	0.872	0.922	5.73%	0.901	0.942	4.55%
Histidine	0.481	0.723	50.3%	0.678	0.813	19.9%	0.824	0.889	7.89%	0.862	0.915	6.15%
Isoleucine	0.455	0.700	53.9%	0.656	0.789	20.3%	0.797	0.870	9.16%	0.839	0.900	7.27%
Leucine	0.474	0.700	47.7%	0.664	0.792	19.3%	0.817	0.881	7.83%	0.856	0.910	6.31%
Lysine	0.684	0.848	24.0%	0.799	0.890	11.4%	0.867	0.923	6.46%	0.881	0.934	6.02%
Methionine	0.700	0.850	21.4%	0.843	0.917	8.78%	0.920	0.957	4.02%	0.948	0.970	2.32%
Phenylalanine	0.509	0.720	41.5%	0.682	0.803	17.7%	0.820	0.885	7.93%	0.860	0.914	6.28%
Threonine	0.394	0.681	72.8%	0.588	0.758	28.9%	0.733	0.836	14.1%	0.767	0.861	12.3%
Valine	0.403	0.664	64.8%	0.613	0.758	23.7%	0.774	0.849	9.69%	0.813	0.878	8.00%
Alanine	0.465	0.699	50.3%	0.656	0.788	20.1%	0.801	0.870	8.61%	0.831	0.895	7.70%
Aspartate	0.429	0.698	62.7%	0.643	0.787	22.4%	0.776	0.861	11.0%	0.817	0.888	8.69%
Glutamate	0.554	0.762	37.6%	0.731	0.844	15.5%	0.851	0.907	6.58%	0.884	0.929	5.09%
Glycine	0.390	0.675	73.1%	0.587	0.756	28.8%	0.745	0.839	12.6%	0.787	0.868	10.3%
Proline	0.415	0.685	65.1%	0.621	0.778	25.3%	0.793	0.869	9.58%	0.837	0.898	7.29%
Serine	0.455	0.704	54.7%	0.639	0.783	22.5%	0.776	0.861	11.0%	0.819	0.890	8.67%
Tyrosine	0.277	0.605	118%	0.800	0.712	42.4%	0.717	0.830	15.8%	0.778	0.875	12.5%
Mean	0.481	0.720	49.7%	0.667	0.802	20.2%	0.805	0.878	9.07%	0.843	0.904	7.24%

**Table 4 animals-13-00603-t004:** Matrix values assigned to exogenous phytase (1000 FTU/kg) in Moss et al. [136].

Item (g/kg)	Starter Diet 1 to 10 Days	Grower Diet 11 to 21 Days	Finisher Diet 22 to 35 Days	Withdrawal Diet 36 to 42 Days
Phosphorus	1.97	1.97	1.97	1.97
Calcium	2.09	2.09	2.09	2.09
Sodium	0.43	0.43	0.43	0.43
Crude protein	6.04	5.29	4.59	4.59
Lysine	0.34	0.30	0.23	0.26
Methionine	0.10	0.09	0.07	0.08
Cysteine	0.19	0.16	0.14	0.15
Threonine	0.28	0.25	0.19	0.22
Isoleucine	0.27	0.22	0.19	0.19
Leucine	0.59	0.46	0.44	0.44
Tryptophan	0.08	0.07	0.06	0.07
Proline	0.41	0.32	0.26	0.26
Serine	0.37	0.29	0.24	0.24
Valine	0.34	0.27	0.24	0.24
Arginine	0.27	0.21	0.18	0.18
Energy (MJ/kg)	0.29	0.28	0.27	0.28

**Table 5 animals-13-00603-t005:** Estimated impacts of phytase (1000 FTU/kg) matrix values on feed ingredient costs. Adapted from Moss et al. [136].

Diet	Control Diet (AUD/tonne)	Phytase Diet (AUD/tonne)	Reduction (AUD/tonne)	Proportion of Intake (%)
Starter (1 to 10 days)	648.14	572.15	76.02	6.0
Grower (11 to 21 days)	632.87	558.56	74.31	18.4
Finisher (22 to 35 days)	603.28	539.68	63.60	45.0
Withdrawal (36 to 42 days)	572.31	511.46	60.85	30.6
Weighted mean	601.94	536.74	65.47	

## Data Availability

This review is based on data published in peer-reviewed journals many of which are open access.

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
