# Peer review of "The Contribution of Phytate-Degrading Enzymes to Chicken-Meat Production"

_animals, 2023, doi:10.3390/ani13040603_

Round 1

Reviewer 1 Report

This paper seems to be a good content as it informs the relationship between Phytase and Phytate.

It's been a long time since I've explained the details in relation to Phytase.

However, since there are many types of phytase and they are supplied at various levels, examples of this seem to be lacking.

Author Response

Reviewer 1

However, since there are many types of phytase and they are supplied at various levels, examples of this seem to be lacking.

Thank you for your comments. Agreed, there are many phytase preparations available on the market; in broad terms, they are of either fungal or bacterial origin. We were careful to make this distinction in our review as the more recent bacterial phytases are superior. However, we deliberately refrained from specifying brand names as we feel this would have been highly inappropriate.     

Reviewer 2 Report

Dear Author 

 Major issue

The manuscript “The contribution of phytate-degrading enzymes to chicken meat production” is well written and organized. However, the paper simply reports a well-established picture that has emerged from such papers over the past years.

Minor grammatical errors.

 Ln. 40: Rewrite this segment to read “ since their introduction in 1991…”

Ln. 160: “may advantage ???...rewrite to make more sense

Ln 181:  delete “hotly”

Ln 189: Delete “of”

Ln 190: Use a full description of …””in Ravinran et al???? study? …

Ln 210: “Increases??” or increased

Ln 339:  Rewrite this sentence it is not very clear

Ln 467: In the Amerah et al???- rewrite this sentence to make sense

Ln 522. Is the subtitle supposed to be “effect of phytase on energy metabolism”… if so reword the title

Ln 567: Delete “A” to read “……by an average of $65.47 per tonne

Author Response

Reviewer 2

The manuscript “The contribution of phytate-degrading enzymes to chicken meat production” is well written and organized. However, the paper simply reports a well-established picture that has emerged from such papers over the past years.

Thank you for your comments. However, we would contend that our consideration of both Pept-1 and NHE in relation to the impact of phytases on amino acid digestibilities is entirely novel and certainly not part of a “well-established picture”

 Minor grammatical errors.

 Ln. 40: Rewrite this segment to read “ since their introduction in 1991…” Complied

Ln. 160: “may advantage ???...rewrite to make more sense Complied

Ln 181:  delete “hotly” Complied

Ln 189: Delete “of” Complied

Ln 190: Use a full description of …””in Ravinran et al???? study? … Complied

Ln 210: “Increases??” or increased Increases is correct

Ln 339:  Rewrite this sentence it is not very clear Sentence has been re-written

Ln 467: In the Amerah et al???- rewrite this sentence to make sense Sentence has been re-written

Ln 522. Is the subtitle supposed to be “effect of phytase on energy metabolism”… if so reword the title Subtitle has be re-worded

Ln 567: Delete “A” to read “……by an average of $65.47 per tonne Changed A to Aus as the 65.47 are Australian dollars

Reviewer 3 Report

Thank you very much for the work sent to me for review. In general, I have no objections to its preparation, but would the authors not consider introducing a separate chapter on the negative impact of phytate-degrading enzymes on poultry health.

Author Response

Reviewer 3

Thank you very much for the work sent to me for review. In general, I have no objections to its preparation, but would the authors not consider introducing a separate chapter on the negative impact of phytate-degrading enzymes on poultry health.

Thank you for your comments. We feel rather strongly that a section on negative impact of phytate-degrading enzymes on poultry health is simply not justified. At one point phytases were associated with the problems arising from ‘wet litter’, but these associations we not justified.

Reviewer 4 Report

The paper “animals-2123730” provides an extensive review regarding the role of exogenous phytases in improving the sustainability in chicken meat production, with special emphasis on the physiological mechanisms explaining the improvement of nutrients digestibility and utilization when phytases are included in broiler diets.

In my opinion the paper is well organised and well written.

I have just a main comment regarding the contents of the review. The authors correctly underlined that the use of exogenous phytases plays a key role for the sustainability of poultry production either from environmental or economical point of view. However, these enzymes are crucial also for the gut health of the animals. Improving gut health avoiding the use of antibiotics is a key contributor for the sustainability of the poultry sector. Thus, my question is: does the review deserve a deepening on the relationship between phytases and health of broiler chickens?

Minor comments are listed below.

L22: The use of the symbol “(P)” in the abstract seems not necessary. The sentence could be changed this way: “The genesis of this remarkable development is based on the capacity of phytases to enhance phosphorus utilisation, thereby reducing its excretion.”

L24: “IP6” is not necessary, it can be deleted

L29: Delete “(P)”

L36: the abbreviation “NHE” is not used in the abstract. Please delete

L123: The acronym “AMEn” should be specified

L124: Please specify the place you mean with “this Campus”

L125: “in improving weight gain…”

L127 “improved FCR (2.53%) in broiler chickens”

L132 “IP6

L165: change “depressed” to “worsened”

Table 1: The inclusion rate of phytase should be reported for the studies analysed by Cowieson et al. [48]. Perhaps, you can use different indicators to separate different levels of inclusion and clarify the meaning of each indicator in tables footnotes.

L282: Delete “in”

L342: The symbol “Na” appeared before (L341) without being specified.

L461: “reported in” can be deleted

L603: Delete “in”

Author Response

Reviewer 4

I have just a main comment regarding the contents of the review. The authors correctly underlined that the use of exogenous phytases plays a key role for the sustainability of poultry production either from environmental or economical point of view. However, these enzymes are crucial also for the gut health of the animals. Improving gut health avoiding the use of antibiotics is a key contributor for the sustainability of the poultry sector. Thus, my question is: does the review deserve a deepening on the relationship between phytases and health of broiler chickens?

Thank you for your comments. Gut integrity in broiler chickens is important, agreed. However, the impacts, if any, of exogenous phytases on the gut microbiota have yet to be defined. Moreover, the importance of the gut microbiota in relation to gut integrity and growth performance had yet to be clarified. We are remined of the statement made by Bindari and Gerber (2021): “However, microbial signatures associated with productivity remain elusive because of the high variability of the microbiota of individual birds resulting in multiple and sometimes contradictory profiles associated with poor or high performance.”

Bindari YR, Gerber PF (2021) Centennial Review: Factors affecting the chicken gastrointestinal microbial composition and their association with gut health and productive performance. Poultry Science 101, 101612

L22: The use of the symbol “(P)” in the abstract seems not necessary. The sentence could be changed this way: “The genesis of this remarkable development is based on the capacity of phytases to enhance phosphorus utilisation, thereby reducing its excretion.” Agreed, changed accordingly

L24: “IP6” is not necessary, it can be deleted Deleted

L29: Delete “(P)” Deleted

L36: the abbreviation “NHE” is not used in the abstract. Please delete Deleted

L123: The acronym “AMEn” should be specified Specified

L124: Please specify the place you mean with “this Campus” The Camden Campus now inserted

L125: “in improving weight gain…” Corrected

L127 “improved FCR (2.53%) in broiler chickens” Corrected

L132 “IP6” Corrected

L165: change “depressed” to “worsened” Changed to “compromised”

Table 1: The inclusion rate of phytase should be reported for the studies analysed by Cowieson et al. [48]. Perhaps, you can use different indicators to separate different levels of inclusion and clarify the meaning of each indicator in tables footnotes. The details are now included in the text. In the meta-analysis phytase inclusion ranged from 500 to 2000 FTU/kg with a weighted average of 872 FTU/kg.

L282: Delete “in” Deleted

L342: The symbol “Na” appeared before (L341) without being specified. Corrected

L461: “reported in” can be deleted Deleted

L603: Delete “in” Deleted